# Fault Detection in Induction Motor Using Time Domain and Spectral Imaging-Based Transfer Learning Approach on Vibration Data

**DOI:** 10.3390/s22218210

**Published:** 2022-10-26

**Authors:** Sajal Misra, Satish Kumar, Sameer Sayyad, Arunkumar Bongale, Priya Jadhav, Ketan Kotecha, Ajith Abraham, Lubna Abdelkareim Gabralla

**Affiliations:** 1Mechanical Engineering, Galgotias College of Engineering and Technology, Dr. A.P.J. Abdul Kalam Technical University, Greater Noida 201306, India; 2Symbiosis Centre for Applied Artificial Intelligence, Symbiosis International (Deemed University), Pune 412115, India; 3Symbiosis Institute of Technology, Symbiosis International (Deemed University), Pune 412115, India; 4Machine Intelligence Research Laboratories, Auburn, WA 98071, USA; 5Department of Computer Science and Information Technology, Princess Nourah bint Abdulrahman University, P.O. Box 84428, Riyadh 11671, Saudi Arabia

**Keywords:** fault diagnosis, induction motor, Short Time Fourier Transform, transfer learning, vibration signal

## Abstract

The induction motor plays a vital role in industrial drive systems due to its robustness and easy maintenance but at the same time, it suffers electrical faults, mainly rotor faults such as broken rotor bars. Early shortcoming identification is needed to lessen support expenses and hinder high costs by using failure detection frameworks that give features extraction and pattern grouping of the issue to distinguish the failure in an induction motor using classification models. In this paper, the open-source dataset of the rotor with the broken bars in a three-phase induction motor available on the IEEE data port is used for fault classification. The study aims at fault identification under various loading conditions on the rotor of an induction motor by performing time, frequency, and time-frequency domain feature extraction. The extracted features are provided to the models to classify between the healthy and faulty rotors. The extracted features from the time and frequency domain give an accuracy of up to 87.52% and 88.58%, respectively, using the Random-Forest (RF) model. Whereas, in time-frequency, the Short Time Fourier Transform (STFT) based spectrograms provide reasonably high accuracy, around 97.67%, using a Convolutional Neural Network (CNN) based fine-tuned transfer learning framework for diagnosing induction motor rotor bar severity under various loading conditions.

## 1. Introduction

Mechanical and electrical tools are broadly utilized in different industrial fields, and their dependability is connected with the industry’s monetary advantages and the well-being of society [1,2]. Mechanical equipment plays a vital role in any industry for its functioning and development. This equipment also requires high capital costs and maintenance costs [3]. Fault diagnosis assumes a critical part in machinery like an induction motor, commonly used in many industrial systems. Induction motors are reliable, but faults can happen because of their heavy-duty cycle and corrosive environment [4]. The faults in induction motors are broadly classified as electrical and mechanical faults [5]. Electrical faults mainly include rotors and stator faults, whereas mechanical faults include bearing and eccentricity faults [6]. In the rotor, faults are caused due to broken rotors or broken end ring faults. Broken rotor bars are one of the problems that frequently arise in an induction motor. An induction motor’s rotor is made up of several bars that, if damaged, might cause the motor to malfunction severely. The broken bar faults initially appear as a localized fracture of the rotor bar with stress increasing nearby. As the fault progresses, the rotor bar breaks completely, including adjacent bars, and eventually the entire motor fails. If there is no monitoring of the machines’ status and maintenance is neglected, the failures that ensue could be very dangerous and expensive. even the minor, initially unimportant fault. The presence of broken rotor bars prompts security concerns, reduced torque generation, and also irregular motor working [7]. Broken bar defects can arise for various reasons including:Magnetic stresses resulting from electromagnetic pressures, magnetic asymmetry forces, sounds, and electromagnetic vibrations;Thermal stresses resulting from over-load, uneven heat distribution, hot spot, and arc;Remaining stress from the manufacturing process;Dynamic stress brought on by the axial torque and centrifugal forces of the rotor;Circumferential stress brought on by the rotor material’s deterioration and contamination by chemical substances and humidity;Mechanical stress brought on by mechanical fatigue of various parts, bearing damage, loosened laminations, etc.

These faults might cause the motor to malfunction, resulting in a drop in efficiency, increased operational costs, and high maintenance costs if not dealt with in time. Small pieces of the broken rotor bars could rise out of the space because of the centrifugal force of the motor’s rotation, which could damage the stator winding and lamination if they come in contact [8]. Also, uneven heating arises in rotor bars due to adjacent broken bars which could generate eccentricity leading to an unbalanced magnetic pull [9], which could also lead to breakage of other rotor bars [10], and due to adjacent broken bars, rated current increases [11], resulting a decrease in the average torque generation [12].

To guarantee the accessibility of the industrial system and the security of items and people on the premises, the monitoring, and detection of rotor failure cannot be overstated. Rotor abnormality in the mechanical equipment can be analyzed from the monitoring data by various processing techniques at the beginning phase. Early fault detection and diagnosis could minimize losses and decrease the chances of machine failure or spreading failure, and limit fault severity. Utilizing a proper strategy for extracting the characteristics that demonstrate the occurrence of rotor failures in the monitored data to improve the prediction accuracy of the motor’s state and the analysis effectiveness [13]. Generally, vibrational data are broadly utilized for fault diagnosis among the different data that could be examined from a three-phase induction motor. The work is structured as follows to classify faults in the induction motor by using the vibration signals:The open-source dataset of the rotor with the broken bars in a three-phase induction motor is available on the IEEE data port and is used to extract the data’s time, frequency, and time-frequency domain features.The extracted time-domain and frequency-domain features are provided to ML models for the rotor fault classification.This study also provides a viable method of applying time-frequency-based feature extraction (STFT) on signals from multiple sensors to convert the one-dimensional time-series signal into a two-dimensional image to learn better information using fine-tuned transfer learning models.

## 2. Related Work

To guarantee the accessibility of the industrial system and the security of items and people on the premises, the monitoring, and detection of rotor failure cannot be overstated. Rotor abnormality in the mechanical equipment can be analyzed from the monitoring data by various processing techniques at the beginning phase.Many researchers used different approaches to diagnose the faults in the induction motor. Ola E. Hassan et al. [5] presented a diagnosis based on different loading conditions, several broken rotor bars, validation methodology, and numerous signal processing methods like FFT, Wigner-Ville distribution, etc., for identifying rotor broken bars. R. Zhang et al. [14] presented a model for bearing failure diagnosis in an induction motor in different working conditions with the assistance of transfer learning by incorporating the use of neural networks, vibration data, and the sliding frame technique to generate training samples from the vibrational data obtained. Pu Shi et al. [15] used a new diagnostic technique based on the wavelet coefficient of stator current to indicate parameters under different loading conditions for different levels of broken rotor bar severity. ZHU et al. [16] presented a new technique by combining the strength of Deep Transfer learning with CNN and Domain adaptation with the help of a domain loss that combines various Gaussian kernels for better feature adaptation. They used the raw vibration data instead of spectrum features by converting the time-domain signal into the image through a defined effective method. 

Yan Du et al. [17] proposed a novel approach for the diagnosis in induction motors by employing the use of STFT for feature extraction from the raw vibrational data and proposing a novel network model with the use of transfer learning, named Transfer Deep Residual network for improving the fault diagnosis performance. The output from the STFT feature extractor was fed into the TDRN as an input for the fault diagnosis process. J. W. Hur et al. [18] presented several analysis techniques for the fault diagnosis in the BLDC motor in non-stationary operating conditions by observing the change in the vibrational data of the motor. They dealt with different vibration signal analysis techniques for anomaly detection in the BLDC motor working under different speed conditions as well as providing different condition indicators for determining the SOH of the system. R. Supangat et al. [19] Instead of using the FFT, which is based on the induction motor’s load condition, a newly developed technique using the wavelet transform was accustomed to exploring the broken rotor bar failure identification using the starting current analysis in an induction motor to identify the typically broken bar fault frequencies from the overshadowing of the basic components. 

Ferrucho-Alvarez, E.R. et al. [20] propose a novel approach for detecting damaged rotor bars in an induction motor based on Contrast estimation, similar to the one used in image processing. Their experimental results from applying the contrast method validate Unser and Tamura contrast definitions as a means of recognizing and characterizing an Induction motor operating state as Healthy, 1 bar broken, or 2 bar broken. S. Mohanty and D. Mohanty [21] investigated the impact of fault signatures caused by damaged bars in an induction motor under varying loading conditions. A new method based on wavelet transform was derived, and simulations were used for diagnosing procedures to diagnose the broken bar fault. N. Mehala and R. Dahiya [22] presented the theoretical and experimental results for identifying faults in induction motors using Motor Current Signature Analysis (MCSA) using the current spectrum obtained from FFT performed on signal under analysis. They also compared different signal processing methods performed under non-constant load conditions in an induction motor.

Almounajjed A. and Sahoo AK. [23] proposes an innovative online detection scheme for diagnosing incipient inter-turn short circuit faults and estimating failure severity in induction motors to provide the motor with a safe operating area by introducing a new mathematical variable based on the discrete wavelet analysis and using a multi-class SVM to carry out the classification function, their work intends to estimate the proportion of faulty turns in the shorted winding. Tauheed Mian et al. [24] examined different bearing failure combinations, including dual and multiple fault circumstances, using two widely used fault diagnosis techniques: non-invasive infrared thermography (IRT) and vibration monitoring in the time-frequency domain by extracting scalograms. They used a CNN network to classify the fault combinations. A. Choudhary et al. [25] reviewed SOTA Condition Monitoring and Fault Diagnosis approaches for early identification of failure in electric vehicles and reducing the chance of abrupt failure. They also discussed difficulties for future work on emerging technologies for research activities. Chang, H.-C et al. [26] showed the usage and advantages of DCGAN in generating the dataset for fault diagnosis to oversample the Imbalanced data. Their results showed promising results when dealing with imbalanced data and using DCGAN and CNN on time-frequency features for fault severity classification tasks.

Based on the available literature on induction motor fault diagnosis, the author used the raw signal and extracted the features using the time-domain, frequency-domain, and time-frequency domain analysis. The time series data was converted into a frequency domain by using the Fast Fourier Transform (FFT) technique. In the time-frequency domain, STFT was used to transform a one-dimensional time-series signal into a two-dimensional image called spectrograms. The spectrograms were fed to the CNN-based fine-tuned transfer learning models for fault classification. 

Along with time-domain and frequency-based feature extraction for diagnosis, the critical contribution of this study is the use of spectrograms for fault identification under various loading conditions generated from the vibration signals of an induction motor and the use of Transfer learning to classify the severity of the fault in less amount of data with improved accuracy.

## 3. Database Description

In this study, the IEEE data port’s dataset “Experimental database for detecting and diagnosing rotor broken bar in a three-phase induction motor” provided by Treml et al. was used to get the proposed working system for diagnosing broken rotor bars in an induction motor [27]. Figure 1 shows block diagram of the experimental setup of the workbench used for generating the dataset [27]. The database carries mechanical and electrical signals that were collected with varying mechanical loads on the three-phase induction motor axis and various degrees of broken bar defects in the induction motor rotor. All signals were captured simultaneously for Eighteen seconds for each loading condition, and ten iterations from transient to consistent conditions of the induction motor were recorded. Tests were carried out on healthy motors and motors with rotor defects in direct start with balanced three-phase supply voltage and 60 Hz frequency.

The database was gathered in the “Laboratory of Intelligent Automation of Processes and Systems and Laboratory of Intelligent Control of Electrical Machines, School of Engineering of São Carlos of the University of São Paulo (USP), Brazil”. The database contains reading from Vibro-control uniaxial accelerometers, model PU 2001, with a sensitivity of 10 mV/mm/s, a frequency range of 5 to 2000 Hz, and stainless-steel housing, which provides the integrated acceleration signal over time i.e., provides the measure of vibration velocity from sensors placed at 5 different positions to get different readings of mechanical vibrations along with voltages and current in phases A, B, and C. The reading was collected on various loading conditions to diagnose the failures for healthy engines and engines with rotors containing 1, 2, 3, and 4 bars broken adjacent. It was essential to drill the rotor to imitate the failure of broken bars in the squirrel cage rotor of the three-phase induction motor. To ensure that the diameter of the hole surpasses the width of a rotor bar, drilling was done using a bench drill outfitted with a 6 mm diameter drill, with the tip centered at half the longitudinal length of the rotor. To create a database of tracked variables, a rotor without a hole i.e., a healthy rotor was tested first, and then it was gradually replaced.

The data set is coordinated as a construction of the MATLAB application. The “struct_rs” structure presents the exploratory information alluding to the defect-less induction motor, “struct_r1b” implies one broken bar, “struct_r2b” suggests two broken bars, “struct_r3b” referring three broken bars and “struct_r4b” for four broken bars. Figure 2 shows the reading containing mechanical and electrical signals collected from different sensors on the induction motor, listing three-phase voltage, three-phase current, and five vibration signals.

## 4. Proposed Work

In this work, the authors consider the vibration signal data from the database available in .mat format (data is stored in MATLAB’s structural array format). The python program is used to retrieve the time-series vibration signals (5 different vibration locations as mentioned in Figure 2) and convert them into CSV formats for building the proposed system. Each CSV file in the extracted dataset contains values of vibration data for different locations and loading conditions. The whole procedure consists of two major parts: feature extraction and fault classification.

### 4.1. Feature Extraction

A proper signal-processing technique must be selected to extract useful data from the raw data [28]. Generally, different feature extraction techniques such as time domain, frequency domain, and time-frequency domain techniques were used for extracting useful information from raw data. In this paper, the author used all three methods, i.e., time, frequency, and time-frequency domain analysis, to extract the features from raw data.

#### 4.1.1. Time Domain Analysis

An examination of physical signals, mathematical functions, or time series of economic or environmental data in the context of time is referred to as a time domain analysis. In discrete time or continuous time, the value of the signal or function is understood for all real numbers at various discrete occurrences. Multiple features like mean, variance, Root Mean Square (RMS), standard deviation, peak-to-peak factor, skewness, and kurtosis can be extracted from the raw data. These values are estimated by segmenting the complete vibration data by each second. Those features are calculated on each one-second window for the whole Eighteen-seconds vibration signal. All the features are concatenated to give a single file containing the time-domain features of a raw vibration signal.

#### 4.1.2. Frequency Domain Analysis

Frequency domain analysis (FDA) is the study of physical signals, mathematical functions, or time series data in the context of frequency. According to a range of frequencies, the frequency domain reveals how much of the signal is contained inside a certain frequency band. A frequency-domain representation can additionally contain the phase shift that must be applied to each sinusoid to recombine the frequency components and retrieve the original time signal. Techniques like Fourier transformation are generally used to transform the time-series signal into the frequency domain. The Fast Fourier Transform is used in frequency domain analysis in this work. The FFT is applied to the raw signal, which then estimates the frequency components of the vibration data. Features are then estimated based on a one-second window from the complete vibration signal data.

##### Fast Fourier Transform

Fast Fourier Transform (FFT) is used to transform the spaced time samples and evaluate the frequency domain response. The Discrete Fourier Transform (DFT) computation takes a long time and requires N^2^ floating-point multiplications. The DFT can be quickly calculated using the FFT technique. The FFT is a group of operations created to cut down on the number of unnecessary computations. Features and benefits vary depending on the FFT’s implementation. The 2-point DFT and 4-point DFT are combined with the 8-point, 16-point, to 2r-point extensions to create the FFT technique. The FFT formula breaks down the DFT equation, as shown in Equation (1):(1)X(k)=∑x(n)WNnk

Into a series of small transformations before recombining them. The FFT algorithm is used to transform a signal (*x*) of length (*N*) from the time domain to the frequency domain (*X*) as shown in Equation (2).
(2)$$X\left[h\right]=\sum_{i=0}^{N-1}\left(x\left[i\right]W_{N}^{ih}\right),W_{N}=e^{-\frac{J2\Pi }{N}},\;\mathrm{for}\;h=0,1,2,\ldots ,N-1\;$$

Figure 3 shows the flowchart of the procedure implemented to extract features in the time domain and frequency domain from the raw vibration data of the three-phase induction motor.

#### 4.1.3. Time-Frequency Domain Analysis

An induction motor is a rotatory machine that generates non-stationary signals whose frequency content changes over time. Brandt expressed that most of the inspection of rotating machines depends on inspecting the vibrational signals [29]. Utilizing frequency domain analysis on this signal would result in transforming the time series signal to the frequency domain. The frequency domain has problems while dealing with non-stationary signals, as it assumes the signal is stationary when it transforms into the frequency domain. Thus, frequency-domain analysis techniques like Fourier Transform (FT) cannot provide time distribution information of the spectral components. The time-frequency domain analysis has been utilized for non-stationary waveform signals, which are generally used when machinery failures occur. Many time-frequency analysis methods have been created and applied to apparatus failure diagnosis, e.g., STFT, wavelet transform (WT), Hilbert-Huang transform (HHT), etc. Figure 4 shows the generalized difference between the time-domain, frequency-domain (FT), and time-frequency domain (STFT) approaches. These approaches indicate that the time domain has good time resolution but no frequency information, whereas the frequency domain has good frequency resolution but poor time resolution. Time-frequency domain provides good time and frequency resolution due to its windowing approach. In this work, the author uses the STFT time-frequency domain approach to transform signals to get information about the spectral components.

##### Short Time Fourier Transform

One of the most popular methods for time-frequency analysis is the STFT, a modified Fourier transform that enables us to analyze non-stationary signals in the time-frequency domain [30]. The concept behind STFT is to utilize a windowed function to extract the windowed signal from the entire signal. It uses a short-time localized window function to divide a signal into equal-length segments. The Fourier spectrum is then extracted using the DFT independently on each windowed signal. The STFT of the continuous-time signal *x*(*t*) can be expressed as follows in Equation (3)
(3)$$stft\left(t,w\right)=\int_{-\infty }^{+\infty }x\left(\tau \right)\omega \left(\tau -t\right)\exp\left(-j\omega \tau \right)d\tau ,\;$$
where ω(τ) is a sliding window function and *τ* is a time variable.

Corresponding, the STFT of a vibration signal in the discrete form *x*(*n*), Equation (4) will become,
(4)$$stft\left(n,w\right)=\sum_{-\infty }^{+\infty }x\left(n\right)\omega \left(n-m\right)\exp\left(-jwn\right)$$

The magnitude squared of the STFT, likewise called the spectrogram, is utilized to approximate the frequency components depending on the STFT features of the signal. The spectrogram is a visual depiction of the signal measuring the variation of frequency as it varies with time. The spectrogram can be presented mathematically by Equation (5),
(5)$$spec\left(n,w\right)=\left|\;stft{\left(n,w\right)}^{2}\right|$$

Figure 5 shows the flowchart of the procedure implemented to generate the spectrograms of the raw vibration data of the three-phase induction motor.

The time and frequency-domain features using FFT extracted at an interval of the 1-s window from multiple sensors are concatenated to form a single file corresponding to one particular label, i.e., the condition of the rotor bar in an induction motor. Each label file containing many feature vectors is finalized as a single file having all the labeled feature rows.

### 4.2. Classification

The function of the classification component is to convert the independent variable into the dependent variable by classifying the features returned by the feature extractor into a group of brain patterns. In this work, some algorithms for classification are K-Nearest neighbors (KNN), Decision tree, and Random Forest, used for data in numerical values, i.e., features from the time domain or frequency domain. For the time-frequency domain, neural networks can be used for more accurate results as the features are converted into two-dimensional images. CNN [31] is most widely used in image classification and performs well in extracting features from the images [32]. They are used in a way that convolution is first conducted using multiple kernels/filters followed by a non-linear activation function, Batch Normalization, and pooling if required. For the classification problem, fully connected layers are used after multiple rounds of convolution, batch normalization, and pooling operations. The parameters of CNN are optimized with backpropagation by minimizing the loss. In this work, CNN-based pre-trained transfer learning models are used for fault diagnosis. Transfer learning is a technique where we use a model trained on a particular task, and remodeled on some other-related task. It uses the knowledge gained by training on a large dataset to apply to small data for better results. Transfer learning is a widely used technique in the area of machine learning or deep learning as it helps to build a model with having small dataset by using someone else’s model, which is trained on a large dataset for a different but similar task and implementing that knowledge to get a more accurate result.

In transfer learning, as shown in Figure 6 [31,33], learning parameters are transferred by the model trained on large data, for a long time on GPUs or TPUs, to our desired model and then freeze the parameters while training our model. Fine-tuning the transfer learning model also helps to achieve the desired result for the particular task. In this work, the CNN operations are conducted with the help of Transfer Learning by using different CNN networks like VGG16, InceptionV3, MobileNetV2, and Inception_ResNetV2. These networks are used to extract features from the spectrogram, and then those features are passed to fully connected networks. Lastly, a Softmax function is applied for the multi-class classification task.

The proposed work steps for broken rotor bar fault diagnosis in an induction motor are summarized as follows:The database in .mat (MATLAB structural array format) was parsed using a python program which goes through each loading condition containing time-series data and all the vibration data in the files are saved in separate .csv files format.A Python program extracts the features from the vibration data from multiple sensors into the time domain and frequency domain (after applying FFT on raw data) by using a window size of 1-s.Another Python program extracts the features from raw data into the time-frequency domain with the help of STFT, and then respective features are converted into spectrograms.The generated spectrograms are then saved in image format and used by the neural networks for fault diagnosis.All the data in multiple domains are then labeled as “rs”, “r1b”, “r2b”, “r3b”, and “r4b” i.e., specific motor conditions.The time domain and frequency domain features are then used to train the ML model for classification. KNN, Decision tree classifier, and Random Forest Classifiers are used for the classification of the broken bar diagnosis.The images of spectrograms are fed into different fine-tuned Convolutional neural networks (CNN) transfer learning models such as VGG16, InceptionV3, MobileNetV2, and Inception_ResNetV2 for feature learning and to build a system that could easily detect the failure in an induction motor due to broken rotor bars with its respective fault labels.

Figure 7 shows spectrograms, i.e., the time-frequency spectrum, of an induction motor vibration data collected from the axial location under normal conditions and conditions with the rotor having healthy, 1, 2, 3, and 4 broken bars adjacent, under loading conditions at 2.5 Nm after applying STFT on each data file. More precise representation can be achieved by reducing the window size in the window function but may result in increased computation time.

The spectrogram generated using STFT at different loading conditions of the vibration data of the three-phase induction motor are stored in a database containing the labels “rs”, “r1b”, “r2b”, “r3b”, and “r4b”. The “rs” indicates a healthy rotor, and the rotor with “r1b”, “r2b”, “r3b”, and “r4b” are broken bars, respectively. Table 1 indicates the number of spectrogram samples generated for each condition and its class label.

Figure 8 shows the working procedure of the proposed work using time-frequency domain analysis and fine-tuned transfer learning model implementation to distinguish the fault detected in the induction motor.

## 5. Results

The rotor fault detection of a three-phase squirrel cage induction motor has two major phases; feature extraction from raw data and fault identification from extracted features using classification models. KNN, Decision Tree, and Random forest classifier were used for fault identification from the time and frequency features.

Table 2 shows the test accuracy of the classification model trained on the features extracted from the time and frequency domain. The confusion matrix portrays the classification ability of the model in the shape of true versus projected value. Figure 9 shows the confusion matrices for the time and frequency domains, respectively.

In time-frequency domain spectrograms, CNN-based pre-trained model structures were selected for classification purposes because of their great strength in learning the 2D image data. For fault identification, spectrograms were generated from the extracted STFT features. All the architectures were trained on the generated spectrograms to test the adaptability of models for the various failure condition under different loading conditions. All the architecture was given the same values of hyper-parameters with the same number of fully connected layers.

Additionally, the predictive outputs from the confusion matrix and classification report of the final layer of the working model for the target data in each situation are given. Figure 10 shows the training and validation graphs generated from the deep-learning model, demonstrating the performance of different architectures used for fault diagnosis.

Table 3 shows the Performance measures and comparison of the CNN network and different transfer learning networks to perform better learning of the fault classification of rotors. The CNN shows the test accuracy up to 89.99%. The pre-trained CNN transfer learning approach models show better classification compared to the normal CNN model. The MobileNetV2 shows the maximum test accuracy of up to 97.67% using the Transfer learning approach. Figure 11 shows the confusion matrix plotted for all the transfer learning networks used for fault diagnosis in an induction motor using time-frequency domain analysis.

The Confusion matrices show that the model easily identifies between spectrograms of various fault signatures under different loading conditions and provides much more accurate results as compared to using time and frequency features. Furthermore, the classification reports describe the classification metrics on an each-class basis. It is a broadly used performance metric in deep learning. The result shows that the transfer learning approach provides the acceptable classification of normal and faulty rotors. Table 4 shows the detailed classification reports of every CNN-based transfer learning network used for learning and mapping the spectrogram images to their corresponding labels.

It can be observed from the confusion matrices and the accuracy report that the time-frequency analysis method (STFT) shows promising results with the help of fine-tuned CNN-based transfer learning models, as most of the labels were classified correctly by every network using the STFT spectrograms as input. The classification accuracy is better for time-frequency domain features than time and frequency domain features.

## 6. Conclusions

This work introduced a viable method of diagnosing induction motor’s rotor fault detection technique under various loading conditions from Vibration signals by performing time-domain, frequency-domain, and time-frequency domain analysis using Short Time Fourier Transform (STFT) and also utilizes the advantages of the Transfer learning approach to get improved accuracy. The time-domain and frequency-domain features provide the test accuracy of up to 87.52% and 88.58%, respectively, using the Random Forest model. In the time-frequency domain, the spectrograms are generated using extracted STFT data of vibration signals, and Fine-tuned CNN framework-based transfer learning models were used for the rotor fault severity diagnosis.

The suggested strategy utilizes the optimal capacity of STFT to process the non-stationary vibration signals by transforming the STFT features into a spectrogram. The fault diagnosis test accuracy obtained are around 95.33%, 94%, 92.33%, and 97.67% for VGG19, InceptionV3, Inception_ResNetV2, and MobileNetV2 models respectively.The use of fine-tuned CNN-based transfer learning models helped to achieve accurate results with having less training data of spectrograms generated under different loading conditionsThe empirical results show that the suggested approach can altogether work in diagnosing the damaged rotor bars fault in a three-phase induction motor with promising results for fault classification in induction motors under different loading conditionsThe method can be used to classify the fault severity by observing the vibration signal from the induction motor and transforming those signals into the time-frequency domain to feed into the model for the classification task.Further improvement can be made by generating and oversampling the database using Generative Adversarial Networks (GANs) and combining that with the transfer learning approach. Furthermore, signal processing techniques like noise filtering or others can be employed to the raw signal to extract more refined features.

## Figures and Tables

**Figure 1 sensors-22-08210-f001:**
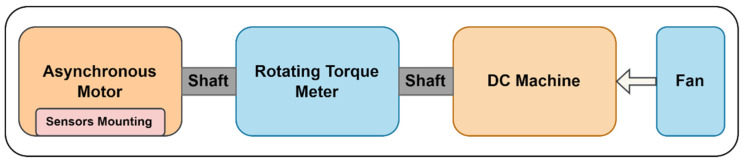
The block diagram of experimental workbench for collecting and generating a database.

**Figure 2 sensors-22-08210-f002:**
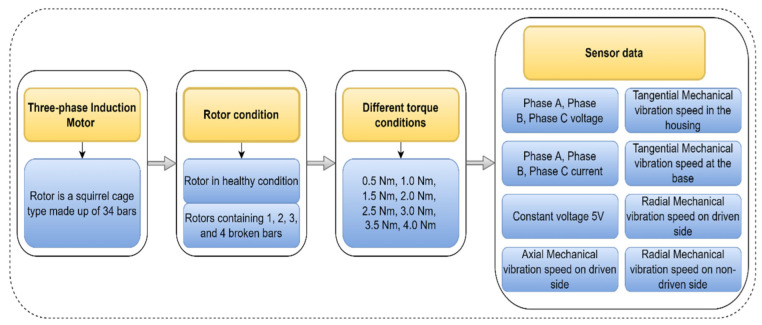
Dataset description of rotor broken bar in a three-phase induction motor.

**Figure 3 sensors-22-08210-f003:**
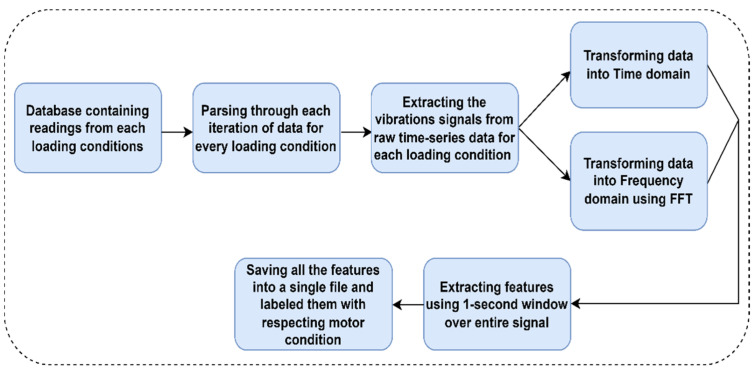
Steps involved in feature extraction in the time domain and frequency domain from raw data.

**Figure 4 sensors-22-08210-f004:**
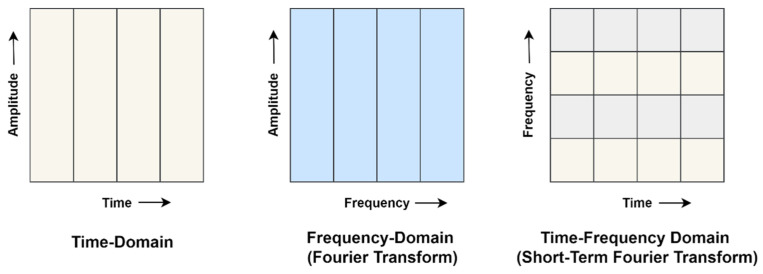
Time, frequency (FT), and time-frequency domain (STFT) signal analysis approach.

**Figure 5 sensors-22-08210-f005:**
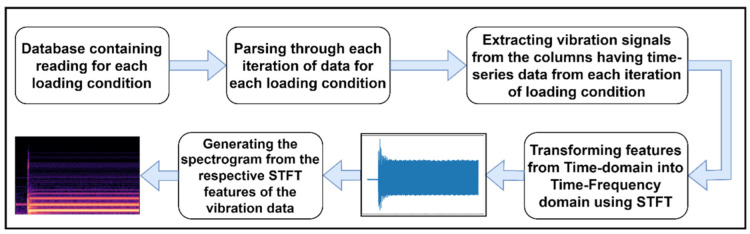
Steps involved in spectrogram generation from raw data.

**Figure 6 sensors-22-08210-f006:**
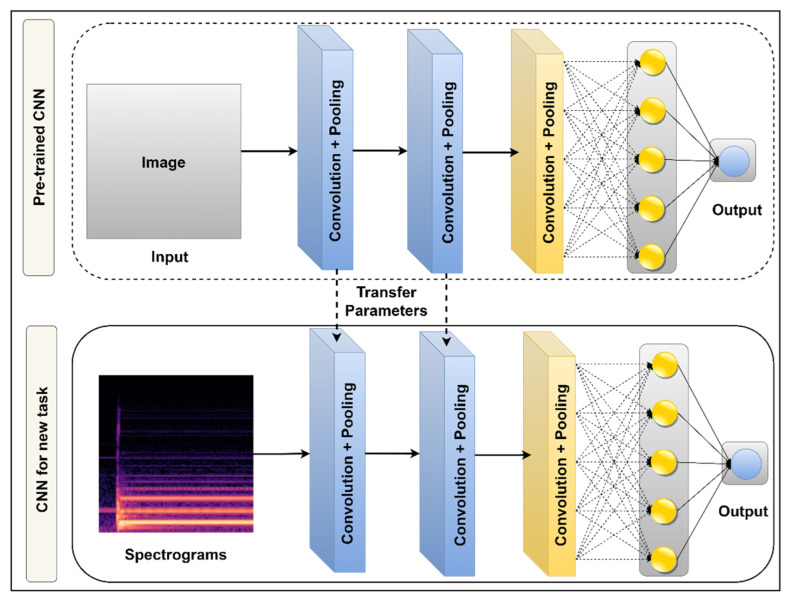
Fine-tuned Transfer Learning approach in pre-trained CNN models.

**Figure 7 sensors-22-08210-f007:**
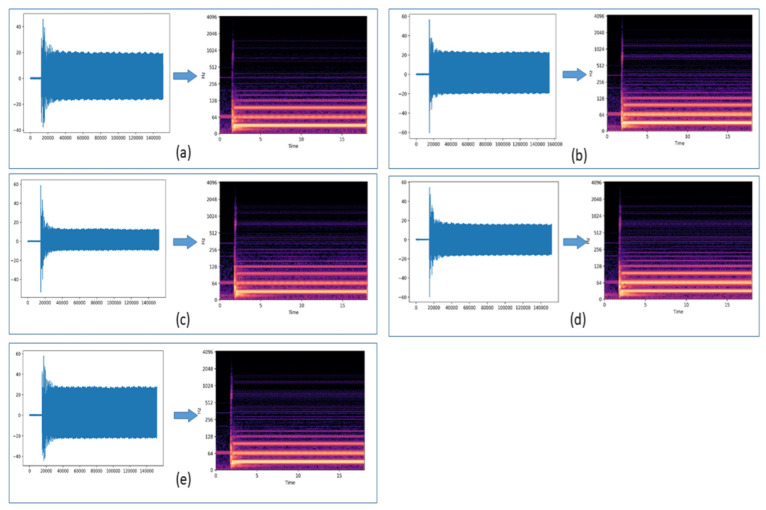
The STFT data plotted and the generated spectrogram of the vibration data for 2.5 Nm torque condition (**a**) healthy rotor (**b**) one broken rotor bar (**c**) two broken rotor bars (**d**) three broken rotor bars (**e**) four broken rotor bar.

**Figure 8 sensors-22-08210-f008:**
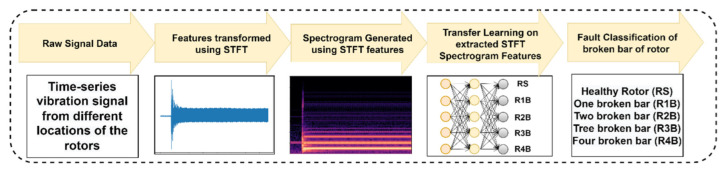
Working procedure of the proposed work using time-frequency domain (STFT) analysis.

**Figure 9 sensors-22-08210-f009:**
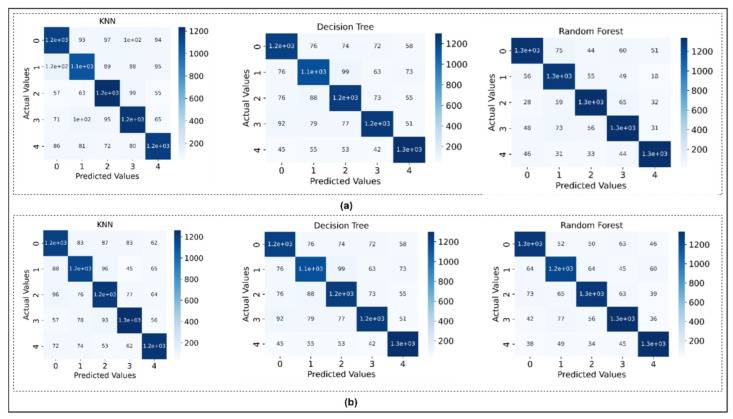
Confusion matrix plotted for models trained on (**a**) time domain features and, (**b**) frequency domain features.

**Figure 10 sensors-22-08210-f010:**
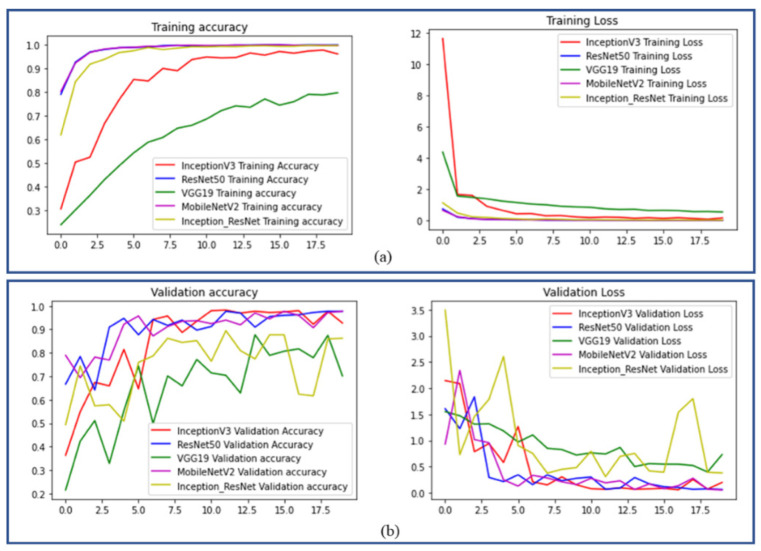
Training and validation graphs of the proposed work with different fine-tuned CNN transfer learning networks (**a**) training accuracy and loss curves (**b**) validation accuracy and loss curves.

**Figure 11 sensors-22-08210-f011:**
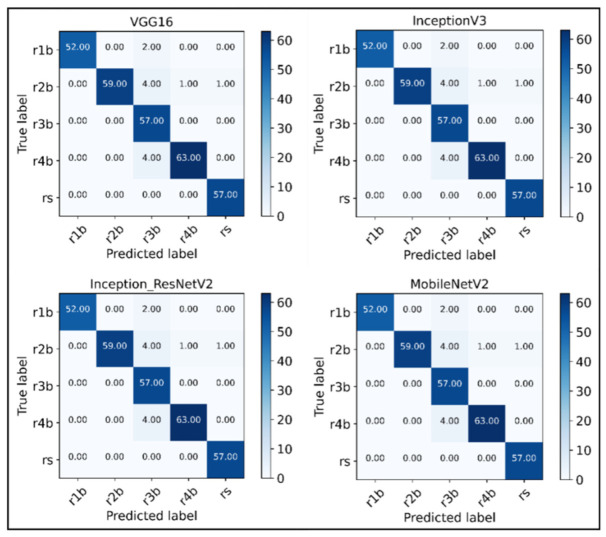
Confusion matrices for the fined-tuned transfer learning networks.

**Table 1 sensors-22-08210-t001:** Number of spectrogram samples generated for each label.

Condition of Rotor	Class Label	Number of Spectrograms Generated
Healthy rotor bar (rs)	0	400
1 broken bar (r1b)	1	395
2 broken bars (r2b)	2	400
3 broken bars (r3b)	3	400
4 broken bars (r4b)	4	400

**Table 2 sensors-22-08210-t002:** Performance measures and comparison of different algorithms for time domain and frequency domain features.

Classification Models	Test Accuracy (%)	
	Using Time-Domain Features	Using Frequency-Domain Features
K-Nearest neighbors	77.37	80.53
Decision Tree	83.80	81.71
Random Forest	86.80	85.92

**Table 3 sensors-22-08210-t003:** Performance measures and comparison of different transfer learning networks.

Networks	Training Accuracy (%)	Validation Accuracy (%)	Test Accuracy (%)
CNN	98.83	91.30	89.99
VGG16	95.89	93.98	95.33
InceptionV3	99.93	94.31	94.00
Inception_ResNetV2	99.93	91.30	92.33
MobileNetV2	99.93	95.32	97.67

**Table 4 sensors-22-08210-t004:** Show the classification reports of transfer learning CNN network used.

Classification report of VGG16 model	Condition	Precision	Recall	F1-score	Support
	1 broken bar (r1b)	0.99	0.91	0.95	54
	2 broken bar (r2b)	0.99	0.92	0.96	65
	3 broken bar (r3b)	0.85	1.00	0.92	57
	4 broken bar (r4b)	0.98	0.94	0.96	67
	Healthy rotor (rs)	0.95	0.98	0.97	57
	Avg. Accuracy			0.95	
Classification report of InceptionV3 model	1 broken bar (r1b)	0.91	0.91	0.91	54
	2 broken bar (r2b)	0.97	0.94	0.95	65
	3 broken bar (r3b)	0.96	0.96	0.96	57
	4 broken bar (r4b)	0.93	0.93	0.93	67
	Healthy rotor (rs)	0.93	0.96	0.95	57
	Avg. Accuracy			0.94	
Classification report of Inception_ResNetV2 model	1 broken bar (r1b)	0.85	0.93	0.88	54
	2 broken bar (r2b)	0.97	0.91	0.94	65
	3 broken bar (r3b)	0.86	0.99	0.93	57
	4 broken bar (r4b)	0.97	0.84	0.90	67
	Healthy rotor (rs)	0.98	0.96	0.97	57
	Avg. Accuracy			0.92	
Classification report of MobileNetV2 model	1 broken bar (r1b)	0.95	0.98	0.96	54
	2 broken bar (r2b)	0.95	0.95	0.95	65
	3 broken bar (r3b)	0.99	0.96	0.98	57
	4 broken bar (r4b)	0.99	0.99	0.99	67
	Healthy rotor (rs)	0.99	0.99	0.99	57
	Avg. Accuracy			0.98	

## Data Availability

Not applicable.

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
