# Peer review of "Fault Detection in Induction Motor Using Time Domain and Spectral Imaging-Based Transfer Learning Approach on Vibration Data"

_sensors, 2022, doi:10.3390/s22218210_

Round 1
Reviewer 1 Report
1. The main contribution of the paper is not clear, please clarify.
2. Please explain how the threshold is obtained in experiments. It seems that the threshold is obtained under no-load condition. Will it still work when the transformer is loaded? How the number of loose nuts and pressing stress affect the determination of threshold?
3. The experiment validation was under no-load condition, but the author said that the method has a great significance for the ‘real-time’ detection of AMDT operation. Please clarify.
4. The failure is detected by the characteristic of vibration signal. This method should be included in the title of the paper.
5. In Fig.1, please explain why the flux density is not the same on the top surface and bottom surface of the two large core columns?
6. There are no experimental setup figures.
7. Please check all the numbers of equations, and the equations should be centered vertically, and align right.
8. There are two “Fig.1”.
9. The indentations of some paragraphs are wrong. For example, page 1-right column-line 45 and page 2-right column-line 32. Please make a careful checking of the others.
10. The format of the ‘References’ is not uniform. For example, [1], [7], [13], [15].
Author Response
Responses to the Reviewer 1 Comments
Journal: Sensors
Title of the manuscript: Fault detection in induction motor using time domain and spectral
imaging-based transfer learning approach
Manuscript Number: sensors-1957200
Dear Editor-in-Chief, Regional Editor and Reviewer,
Thank you for all your useful comments and suggestions for revising our manuscript. We have carefully revised the manuscript accordingly. The detailed revisions/corrections that have been undertaken and our detailed point-by-point response to all the comments given by the reviewer and the Editor-in-Chief/Regional Editor are listed below for your kind review.
Reviewer 1
- The main contribution of the paper is not clear, please clarify.
Our Response:
- Thank you very much for your valuable comments.
- We have added the main contribution of the paper to the manuscript.(Section 2 (Line 160-166)).
---------------------------------------------------------------------------------------
- Please explain how the threshold is obtained in experiments. It seems that the threshold is obtained under no-load condition. Will it still work when the transformer is loaded? How the number of loose nuts and pressing stress affect the determination of threshold?
Our Response:
- Thank you very much for your valuable comments.
- The author uses the online dataset for rotor fault detection.
- It will work under no-load and loading conditions, as the fault classification is based on the training data and generated spectrograms under different conditions.
- The experiment is validated using the spectrograms under different loading conditions with different severity of the broken bars.
- As the data is provided with fault labels, the supervised learning approach is used for fault classification instead of finding the threshold values for each fault condition.
- Experimental condition details are provided on the IEEE Dataport site (https://ieee-dataport.org/open-access/experimental-database-detecting-and-diagnosing-rotor-broken-bar-three-phase-induction)
---------------------------------------------------------------------------------------
- The experiment validation was under no-load condition, but the author said that the method has a great significance for the 'real-time' detection of AMDT operation. Please clarify.
Our Response:
- Thank you very much for your valuable comment.
- As the data is provided with fault labels, the supervised learning approach is used for fault classification instead of finding the threshold values for each fault condition.
- Once the model is trained with the different normal and faulty conditions, it can be used for real-time fault detection under the same experimental conditions.
--------------------------------------------------------------------------------------
- The failure is detected by the characteristic of vibration signal. This method should be included in the title of the paper.
Our Response:
- Thank you very much for your valuable comment.
- The title of the manuscript is updated as per the reviewer's suggestion.
- Updated Title "Fault detection in induction motor using time domain and spectral imaging-based transfer learning approach on vibration data."
---------------------------------------------------------------------------------------
- In Fig.1, please explain why the flux density is not the same on the top surface and bottom surface of the two large core columns?
Our Response:
- Thank you very much for your valuable comment.
- We are unable to understand the reviewer's concern. Please elaborate on the comment so we can able to respond your concern.
--------------------------------------------------------------------------------------
- There are no experimental setup figures.
Our Response:
- Thank you very much for your valuable comment.
- We have updated the draft with Figure 1 as the workbench setup provided by the original authors on the IEEE data port.
--------------------------------------------------------------------------------------
- Please check all the numbers of equations, and the equations should be centered vertically, and align right.
Our Response:
- Thank you very much for your valuable comment.
- We have rechecked the correct order and number of equations and updated the draft with the correct alignment.
---------------------------------------------------------------------------------------
- There are two "Fig.1".
Our Response:
- Thank you very much for your valuable comment.
- We have updated the draft with the corrected figure numbers.
---------------------------------------------------------------------------------------
- The indentations of some paragraphs are wrong. For example, page 1-right column-line 45 and page 2-right column-line 32. Please make a careful checking of the others
Our Response:
- Thank you very much for your valuable comment.
- We have updated the draft with the correct alignment.
--------------------------------------------------------------------------------------
- The format of the 'References' is not uniform. For example, [1], [7], [13], [15].
Our Response:
- Thank you very much for your valuable comment.
- We have crossed check all references and updated the format of the references as per the reviewer's suggestion.
Reviewer 2 Report
The work is good for diagnosing the faults in the Induction motor, by applying the time domain and spectral imaging-based transfer learning approach. I suggest that the manuscript be accepted after some revisions due to the following reasons.
1. Abstract: Abstract should be written in coherent and fluent language due to some essential points such as research purposes, research methods, research contents and research effects, otherwise the innovation and necessity of the manuscript will not be reflected effectively. But the words such as "Purpose", "Methods", " Results and discussion" and "Conclusions" cannot appear in the section. Therefore, the Abstract should have one sentence per each: context and background, motivation, hypothesis, methods, results, conclusions." for example, Reconfigurable Flight Array is repeated five times in the abstract.
2. Introduction: The introduction is too short please add the related literature which is recently been published in this domain. Nonetheless, more passive voice should be used to replace the active voice in sentences. There are some punctuation errors in this section, please correct them.
3. Literature review is week, The first paragraph introducing the research topic may present a much broad and comprehensive view of the problems related to your topic with citations to SOTA references for solving the same problem. SOTA methods can be included in the comparative studies too. Some of the recent work on fault diagnosis on electric motor and its components are as follow:
https://ieeexplore.ieee.org/abstract/document/9900355,
https://www.mdpi.com/2076-3417/12/8/4080,
https://journals.sagepub.com/doi/full/10.1177/01423312221109725?casa_token=eQlglNEUvbwAAAAA%3ATekEs6j1b4nvxN76KUtMqXWTJ27gjTMAwDzxLtPIMth4XF11QHa-Jr7C6vq31mMe_nV13sCcwUmbu2U,
https://www.tandfonline.com/doi/abs/10.1080/10589759.2022.2118747,
https://ieeexplore.ieee.org/abstract/document/9772991,
https://www.mdpi.com/2072-666X/13/5/663
https://www.mdpi.com/1996-1073/15/3/1244,
4. There are some punctuation errors in this section, please correct them.
5. Experimental setup: This section introduces the experimental materials and methods. The pictures are good, but they could be better.
6. 4.2 classification section and proposed Fine-tuned Transfer Learning approach in pre-trained CNN models., some improvements are proposed for the proposed network, they are not well represented in the figure.
7. There is less discussion about the faults, how these faults are created, more details should be added regarding the sensor faults
8. How are the results of the proposed model statistically significant?
9. I suggest to use the bullet points to show your concluding points in the conclusion.
Author Response
Responses to the Reviewer 2 Comments
Journal: Sensors
Title of the manuscript: Fault detection in induction motor using time domain and spectral
imaging-based transfer learning approach
Manuscript Number: sensors-1957200
Dear Editor-in-Chief, Regional Editor and Reviewer,
Thank you for all your useful comments and suggestions for revising our manuscript. We have carefully revised the manuscript accordingly. The detailed revisions/corrections that have been undertaken and our detailed point-by-point response to all the comments given by the reviewer and the Editor-in-Chief/Regional Editor are listed below for your kind review.
Reviewer 2
- Abstract: Abstract should be written in coherent and fluent language due to some essential points such as research purposes, research methods, research contents and research effects, otherwise the innovation and necessity of the manuscript will not be reflected effectively. But the words such as "Purpose", "Methods", " Results and discussion" and "Conclusions" cannot appear in the section. Therefore, the Abstract should have one sentence per each: context and background, motivation, hypothesis, methods, results, conclusions." for example, Reconfigurable Flight Array is repeated five times in the abstract.
Our Response:
- Thank you very much for your valuable comment.
- We have updated the Abstract by keeping the points mentioned.
--------------------------------------------------------------------------------------
- Introduction: The introduction is too short please add the related literature which is recently been published in this domain. Nonetheless, more passive voice should be used to replace the active voice in sentences. There are some punctuation errors in this section, please correct them.
Response: I have updated the Introduction by keeping the points mentioned.
Our Response:
- Thank you very much for your valuable comment.
- We have updated the Introduction by keeping the points mentioned (Section 1 (1. Introduction), Line 46-81)
--------------------------------------------------------------------------------------
- Literature review is week, The first paragraph introducing the research topic may present a much broad and comprehensive view of the problems related to your topic with citations to SOTA references for solving the same problem. SOTA methods can be included in the comparative studies too. Some of the recent work on fault diagnosis on electric motor and its components are as follow:
- https://ieeexplore.ieee.org/abstract/document/9900355,
- https://www.mdpi.com/2076-3417/12/8/4080,
- https://journals.sagepub.com/doi/full/10.1177/01423312221109725?casa_token=eQlglNEUvbwAAAAA%3ATekEs6j1b4nvxN76KUtMqXWTJ27gjTMAwDzxLtPIMth4XF11QHa-Jr7C6vq31mMe_nV13sCcwUmbu2U,
- https://www.tandfonline.com/doi/abs/10.1080/10589759.2022.2118747,
- https://ieeexplore.ieee.org/abstract/document/9772991,
- https://www.mdpi.com/2072-666X/13/5/663
- https://www.mdpi.com/1996-1073/15/3/1244,
Our Response:
- Thank you very much for your valuable comment.
- We have updated the Literature review section as per reviewers suggestions by considering the recent work on fault diagnosis on electric motor and its components more papers having similar scope. (Section 2 (1. Related work), Line 93-96, Line 139-155)
--------------------------------------------------------------------------------------
- There are some punctuation errors in this section, please correct them.
Our Response:
- Thank you very much for your valuable comment.
- We have taken into account the punctuation errors and corrected them.
--------------------------------------------------------------------------------------
- Experimental setup: This section introduces the experimental materials and methods. The pictures are good, but they could be better.
Our Response:
- Thank you very much for your valuable comment.
- We have updated images in Section 2 (Dataset Description)
- We have also added the actual experimental setup diagram from the online dataset repository in the manuscript (Line 178-190)
--------------------------------------------------------------------------------------
- 4.2 classification section and proposed Fine-tuned Transfer Learning approach in pre-trained CNN models., some improvements are proposed for the proposed network, they are not well represented in the figure.
Our Response:
- Thank you very much for your valuable comment.
- The image is updated as per the reviwer's suggestion by adding the "spectrogram" word for better understanding.
--------------------------------------------------------------------------------------
- There is less discussion about the faults, how these faults are created, more details should be added regarding the sensor faults
Our Response:
- Thank you very much for your valuable comment.
- We have updated the draft by mentioning the above points in the dataset description section (3) (Line 191-197)
--------------------------------------------------------------------------------------
- How are the results of the proposed model statistically significant?
Response:
Our Response:
- Thank you very much for your valuable comment.
- Tables 2, 3 and 4 are used to evaluate the model performance, which shows that model is statistically significant.
---------------------------------------------------------------------------------------
- I suggest to use the bullet points to show your concluding points in the conclusion.
Our Response:
- Thank you very much for your valuable comment.
- We have updated the conclusion section by using bullet points. (Line: 460-478)
-------------------------------------------------------------------------------------
--------------------------------------------------------------------------------------
Reviewer 3 Report
The article under review presents a new method of diagnosing an induction motor's rotor fault proposed by the authors. The research was carried out using MatLab and Python programs. The authors claim a diagnostic accuracy of 92% to 97%.
In general, this is an interesting work, the results of which may be useful to specialists in the field of electrical engineering.
During the review, I drew attention to the following shortcomings and I would like to formulate several recommendations:
- It seems that the question of using the results of the study needs to be clarified, namely, for which power range of the induction motors the proposed developments are applicable.
- I think that the practical application of the results obtained is not clearly described in the work. And also the further direction of development of research is not shown. This should be added to the paper.
- Figures 4, 6, 7 and 9 have small inscriptions that should be enlarged to improve perception.
I congratulate the authors on a job well done and recommend the paper for acceptance after minor revision.
Author Response
Responses to the Reviewer-3 Comments
Journal: Sensors
Title of the manuscript: Fault detection in induction motor using time domain and spectral
imaging-based transfer learning approach
Manuscript Number: sensors-1957200
Dear Editor-in-Chief, Regional Editor and Reviewer,
Thank you for all your useful comments and suggestions for revising our manuscript. We have carefully revised the manuscript accordingly. The detailed revisions/corrections that have been undertaken and our detailed point-by-point response to all the comments were given by the reviewer and the Editor-in-Chief/Regional Editor are listed below for your kind review.
Reviewer 3
- It seems that the question of using the results of the study needs to be clarified, namely, for which power range of the induction motors the proposed developments are applicable.
Our Response:
- Thank you very much for your valuable comment.
- We have updated the draft by keeping the points mentioned, and also the dataset description section (3) has been updated with information regarding the induction motor. (Line: 177-189)
--------------------------------------------------------------------------------------
- I think that the practical application of the results obtained is not clearly described in work. And also the further direction of development of research is not shown. This should be added to the paper.
Our Response:
- Thank you very much for your valuable comment.
- We have updated the draft by keeping the points mentioned in the conclusion section. (Line 458-477)
-------------------------------------------------------------------------------------
- Figures 4, 6, 7 and 9 have small inscriptions that should be enlarged to improve perception.
Our Response:
- Thank you very much for your valuable comment.
- We have updated the figures as per the reviewer's suggestion
--------------------------------------------------------------------------------------
Round 2
Reviewer 1 Report
I have no more comments.